# Structural snapshots of V/A-ATPase reveal the rotary catalytic mechanism of rotary ATPases

J. Kishikawa [1,2,4], A. Nakanishi[1,3,4], A. Nakano[1,4], S. Saeki[1], A. Furuta[1], T. Kato [2], K. Mistuoka [3] & K. Yokoyama [1✉]

V/A-ATPase is a motor protein that shares a common rotary catalytic mechanism with $F_oF_1$ ATP synthase. When powered by ATP hydrolysis, the $V_1$ domain rotates the central rotor against the $A_3B_3$ hexamer, composed of three catalytic AB dimers adopting different conformations ($AB_{open}$, $AB_{semi}$, and $AB_{closed}$). Here, we report the atomic models of 18 catalytic intermediates of the $V_1$ domain of V/A-ATPase under different reaction conditions, determined by single particle cryo-EM. The models reveal that the rotor does not rotate immediately after binding of ATP to the $V_1$. Instead, three events proceed simultaneously with the 120° rotation of the shaft: hydrolysis of ATP in $AB_{semi}$, zipper movement in $AB_{open}$ by the binding ATP, and unzipper movement in $AB_{closed}$ with release of both ADP and $Pi$. This indicates the unidirectional rotation of V/A-ATPase by a ratchet-like mechanism owing to ATP hydrolysis in $AB_{semi}$, rather than the power stroke model proposed previously for $F_1$-ATPase.

[1] Department of Molecular Biosciences, Kyoto Sangyo University, Kamigamo-Motoyama, Kita-Ku, Kyoto 603-8555, Japan. [2] Institute for Protein Research, Osaka University, 3-2 Yamadaoka, Suita, Osaka 565-0871, Japan. [3] Research Center for Ultra-High Voltage Electron Microscopy, Osaka University, 7-1, Mihogaoka, Ibaraki, Osaka 567-0047, Japan. [4] These authors contributed equally: J. Kishikawa, A. Nakanishi, A. Nakano. ✉email: yokoken@cc.kyoto-su.ac.jp

The proton translocation ATPase/synthase family includes F-type enzymes found in eubacteria, mitochondria, and chloroplasts, and the V/A type enzymes found in archaea and some eubacteria[1–5] (Fig. 1A). These ATPases produce the majority of cytosolic ATP from ADP and $Pi$ using energy derived from the transmembrane proton motive force generated by cellular respiration[6]. These ATPases share a common molecular architecture, consisting of a hydrophilic $V_1/F_1$ domain responsible for ATP hydrolysis or synthesis, and a hydrophobic $V_o/F_o$ domain housing a proton translocation channel[7–9]. The chemical reaction (ATP hydrolysis/synthesis) in $V_1/F_1$ is tightly associated with proton movement through $V_o/F_o$ using a rotary catalytic mechanism, where both reactions are coupled by rotation of the central rotor complex relative to the surrounding stator apparatus, which includes the ATPase active hexamer[6,10,11] (Fig. 1B).

The V/A-ATPase from the thermophilic bacterium, *Thermus thermophilus* (*Tth*) is one of the best-characterized ATP synthases[3,12]. The overall architecture and subunit composition of V/A-ATPase is more similar to that of the eukaryotic V-ATPase, rather than F-type ATPase. However, the *Tth* V/A-ATPase has a simpler subunit structure than the eukaryotic V-ATPase and shares the ATP synthase function of F-type ATPase[13] (Fig. 1A). The $V_1$ domain of *Tth* V/A-ATPase ($A_3B_3D_1F_1$) is an ATP-driven rotary motor where the central DF shaft rotates inside the hexameric $A_3B_3$ containing three catalytic sites, each composed of an AB dimer. The $V_o$ domain

($E_2G_2d_1a_1c_{12}$) is composed of stator parts including the *a* subunit and two EG peripheral stalks and the $d_1c_{12}$ rotor complex, which consists of a central rotor complex with the DF subunits of $V_1$[14–16]. When ATP hydrolysis by $A_3B_3$ powers the DF shaft, the reverse rotation of the central rotor complex drives proton translocation in the membrane-embedded $V_o$ domain (Fig. 1B).

According to the binding change mechanism of ATP synthesis[6], the three catalytic sites in ATP synthases are in different conformations but interconvert sequentially between three different conformations as catalysis proceeds. Indeed, our previous structure demonstrated that the $A_3B_3$ hexamer in the V/A-ATPase adopts an asymmetrical structure composed of three different AB dimers, termed open ($AB_{open}$), semi-closed ($AB_{semi}$), and closed ($AB_{closed}$)[16,17].

Experimental studies using specific rotational probes attached to DF revealed that ATP-driven rotation of the central shaft was unidirectionally clockwise when viewed from the $V_1$ side[10]. At low ATP concentrations where ATP binding is rate-limiting, rotation proceeds in steps of 120°, commensurate with the three catalytic sites of AB dimers[18]. When using 40 nm gold beads with almost negligible viscous resistance, $V_1$ also pauses every 120° even at an ATP concentration around $K_m$ without a sign of substeps[19]. These single-molecule experiments on $V_1$ suggest that both catalytic events, ATP hydrolysis and product (ADP and $Pi$) release occur at an individual ATP binding position, and imply the presence of chemo-mechanically stable catalytic intermediates (Fig. 1C and Supplementary Fig. 1).

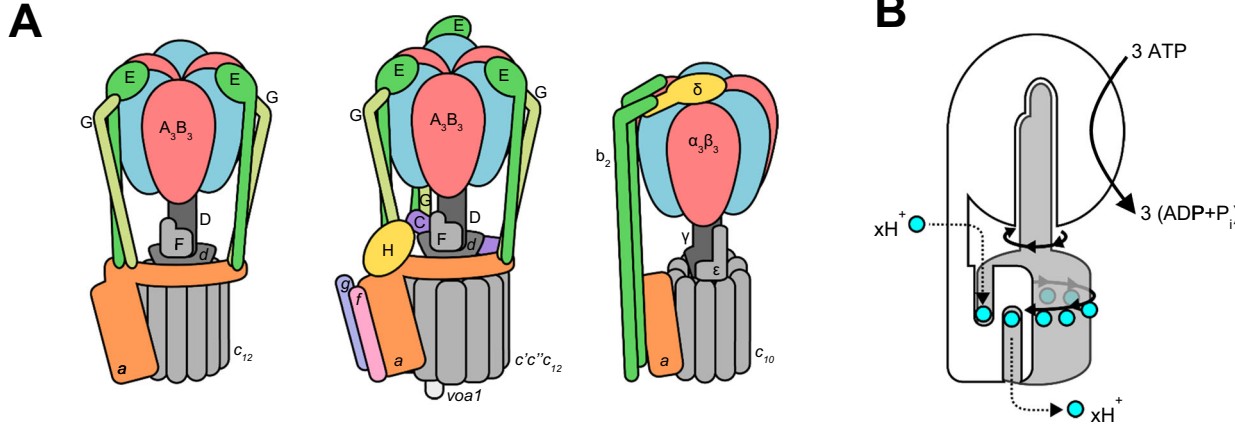

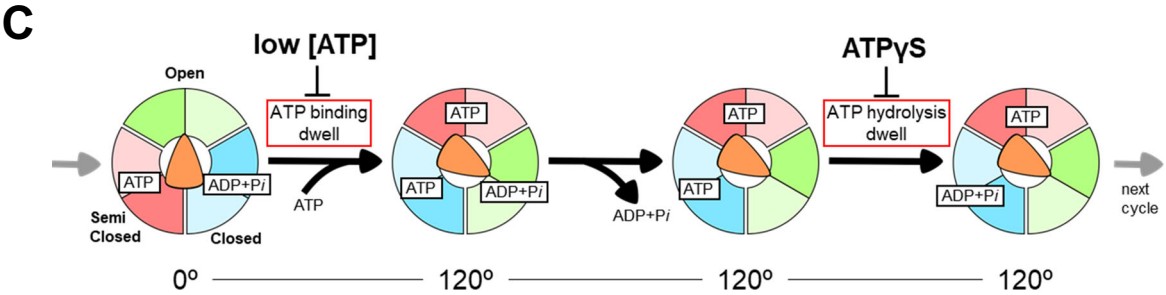

**Fig. 1 Schematic representation of rotary ATPases and the conventional rotary mechanism. A** Illustration of subunit composition of different types of rotary ATPases; prokaryotic V/A-ATPase (*left*), eukaryotic V-ATPase (*middle*), prokaryotic F-ATPase (*right*). The stators are represented in various colors and the rotors are represented in gray. **B** A schematic model of the rotary catalytic mechanism of the V/A-ATPase. When powered by ATP, the central rotor composed of $D_1F_1d_1c_{10}$ (gray) rotates against a surrounding stator composed of $A_3B_3E_2G_2a_1$ (white), coupled with proton translocation across the membrane. **C** The conventional catalytic cycle of V/A-ATPase. At low ATP concentration, the ATP binding dwell time is increased. ATPγS also prolongs the ATP hydrolysis dwell.

However, single-molecule observation experiments only allow us to see the motion of the shaft to which the observation probe is bound, and do not tell us what events are occurring at each catalytic site. To elucidate the entire rotational mechanism of the V/A-ATPase, we must determine the structures of catalytic intermediates of the rotary ATPase during rotation. There are many reaction intermediates of the enzyme during turnover, and this structural heterogeneity makes successful crystallization of a specific state very challenging.

Technological breakthroughs in single-particle Cryo-EM, such as the development of direct electron detectors, and advances in image processing and automation[20,21], have triggered a revolution in structural biology, making this the technique of choice for large and dynamic complexes unsuitable for crystallization. In addition, by freezing Cryo-EM grids at different time points or under different reaction conditions, it is possible to trap intermediate states and thus build up a picture of the chemomechanical cycle of biological macromolecular complexes step by step. To date, there are few examples of studies that have successfully captured such details of a catalytic cycle at atomic resolution using Cryo-EM[22,23].

Here, we report several keys, and thus far uncharacterized, intermediate states of V/A-ATPase, obtained under different reaction conditions. Comparison of these structures provides insight into the cooperativity between the three catalytic sites and demonstrates a rotary catalytic mechanism powered by ATP hydrolysis.

## Results

**Sample preparation for Cryo-EM structural analysis**. We previously determined the Cryo-EM structures of the wild-type V/A-ATPase containing an ADP in the catalytic site of $AB_{closed}$[16,17]. The V/A-ATPase bound to the inhibitory ADP exhibits no ATPase activity until the ADP is removed[13,16,24]. Partial ADP removal from $AB_{closed}$ is possible by dialysis against an EDTA-phosphate buffer, but it is difficult to obtain a homogenous nucleotide-free V/A-ATPase after such a treatment, due to the high binding affinity of the ADP to $AB_{closed}$ (Supplementary Table 1). To obtain a homogeneous ATPase active enzyme, mutant V/A-ATPase (A/S232A, T235S) with reduced nucleotide-binding affinity was purified from *T. thermophilus* membranes[10]. The mutated V/A-ATPase exhibits higher $K_m$ values for nucleotide in both the ATP hydrolysis and synthesis reactions than the wild-type enzyme, but the enzymatic and rotational properties are almost the same as those of the wild-type enzyme[24]. The mutated V/A-ATPase is fully activated for ATPase activity after EDTA/phosphate dialysis; no ADP or ATP was found in the enzyme by quantitative analysis of nucleotides (Supplementary Fig. 2 and Table 1). We incorporated the nucleotide-free V/A-ATPase into nanodiscs comprising the MSP1E3D1 scaffold protein and DMPC. The resulting V/A-ATPase obeys simple Michaelis–Menten kinetics and exhibits ATPase activity of $22 \, s^{-1}$ and the $K_m$ of $394 \, \mu M$ ATP (Supplementary Fig. 2b).

The nucleotide-free V/A-ATPase (Nucfree) was used for cryo-grid preparation under different ATPase reaction conditions (Supplementary Fig. 1c). Results of the structural analysis of the protein under each set of reaction conditions are summarized in Supplementary Fig. 3a–d.

**The structures of V/A-ATPase without nucleotide ($V_{nucfree}$)**. The flow charts showing image acquisition and reconstitution of the 3D structure of V/A-ATPase without nucleotide are summarized in Supplementary Fig. 3a. We obtained structures of three rotational states of V/A-ATPase without nucleotide; state1 at 3.1 Å, state 2 at 4.7 Å, and state 3 at 6.3 Å resolution, with the

DF shaft positions differing by 120° in each case (Fig. 2). Using signal subtraction of the $V_o$ domain, we achieved resolution of 3.0 and 4.1 Å for the $V_1$ domain including half of the EG stalk in state1 and 2, allowing us to build atomic models of the $V_1$ domain of these states.

The three AB dimers in the $V_1$ domain adopted open ($AB_{open}$), semi-closed ($AB_{semi}$), and closed ($AB_{closed}$) states, respectively (Fig. 2B, C). The tip of the C-terminal helix bundle (CHB) of $A_{open}$ is in contact with the C-terminal helix of the D subunit, and the wide part of the CHB of $B_{open}$ is in contact with the N-terminal helix of the D subunit, respectively (Supplementary Fig. 4a–c). The $AB_{semi}$ and $AB_{closed}$ also interact with the coiled-coil of subunit D in specific regions of the CHB, respectively (Supplementary Fig. 4d–g).

The differences in the structures of the three AB dimers, when superimposed on the β barrel domains of both A and B subunits, are the result of the movement of the N-terminal bulge domain, the nucleotide-binding domain (NB) of the A subunit and the CHBs of both the A and B subunits (Fig. 2D). When comparing the structure of $AB_{open}$ and $AB_{semi}$, both the NB and CHB of the $A_{semi}$ are in closer proximity to $B_{semi}$ than $B_{open}$, resulting in a closed structure of $AB_{semi}$ (Fig. 2C, D). The structure of $B_{open}$ is very similar to $B_{semi}$, as shown in Fig. 2D. In the $AB_{closed}$, both the CHB and NB domains of $A_{closed}$ are in closer proximity to $B_{closed}$, and the CHB of $B_{closed}$ moves to $A_{closed}$, resulting in the more closed structure of $AB_{closed}$ compared to $AB_{semi}$ (Fig. 2C, D).

In the $AB_{closed}$ and $AB_{semi}$ dimers, densities for the catalytic side chains are well resolved, but no density corresponding to nucleotide was observed (Fig. 3A). Hereafter we refer to the structure as the $V_{nucfree}$. The structure of $V_{nucfree}$ is very similar to the previously reported ADP inhibited structure[16,17]. For state1, the *rmsd* value for the $C_\alpha$ chains of $A_3B_3DF$ of the $V_{nucfree}$ and ADP inhibited structures is 1.98 Å (Supplementary Fig. 5). In addition, the $V_{nucfree}$ is also similar to the structures under the saturated-ATP condition determined in this study, with the positions of the catalytic side chains almost identical in both cases (Supplementary Fig. 6). This indicates that the $V_1$ domain adopts the same conformation, including the arrangement of the DF shaft in the $A_3B_3$ and the geometry of the catalytic side chains, irrespective of the presence or absence of bound ATP.

In the density maps obtained for state1 of $V_{nucfree}$, the CHB of the AB dimers was slightly blurred, likely due to structural heterogeneity. To classify the probable substates of state1, we performed focused 3D classification using a mask covering $AB_{open}$ and $B_{semi}$ (Supplementary Fig. 3a). We identified a cryoEM structure of the original state1 from 39,902 particles at 3.1 Å resolution and another substate from 24,101 particles at 3.1 Å resolution. We termed the substates reconstructed from these major particle classes as state1-1 and state1–2, respectively. The atomic model initially constructed as state1 is identical to the atomic model of state1–1. The structures of the sub-states are very similar, with most differences due to the movement of the CHB of the A and B subunits. Therefore, we quantified the difference in the structure of the CHB observed when the structures were superimposed on the N-barrel domain (Supplementary Tables 2 and 3). Substates were also obtained under other reaction conditions (see below) and the *rmsd* values shown in Supplementary Tables 2 and 3 are used to discuss which subunits are responsible for the differences in the structure of the substates obtained under different reaction conditions.

**Structures obtained at a saturating ATP concentration**. Cryo-grids were prepared using a reaction mixture of nucleotide-free V/A-ATPase, containing the regenerating system and ATP at a saturating concentration of 6 mM. The reaction mixture was

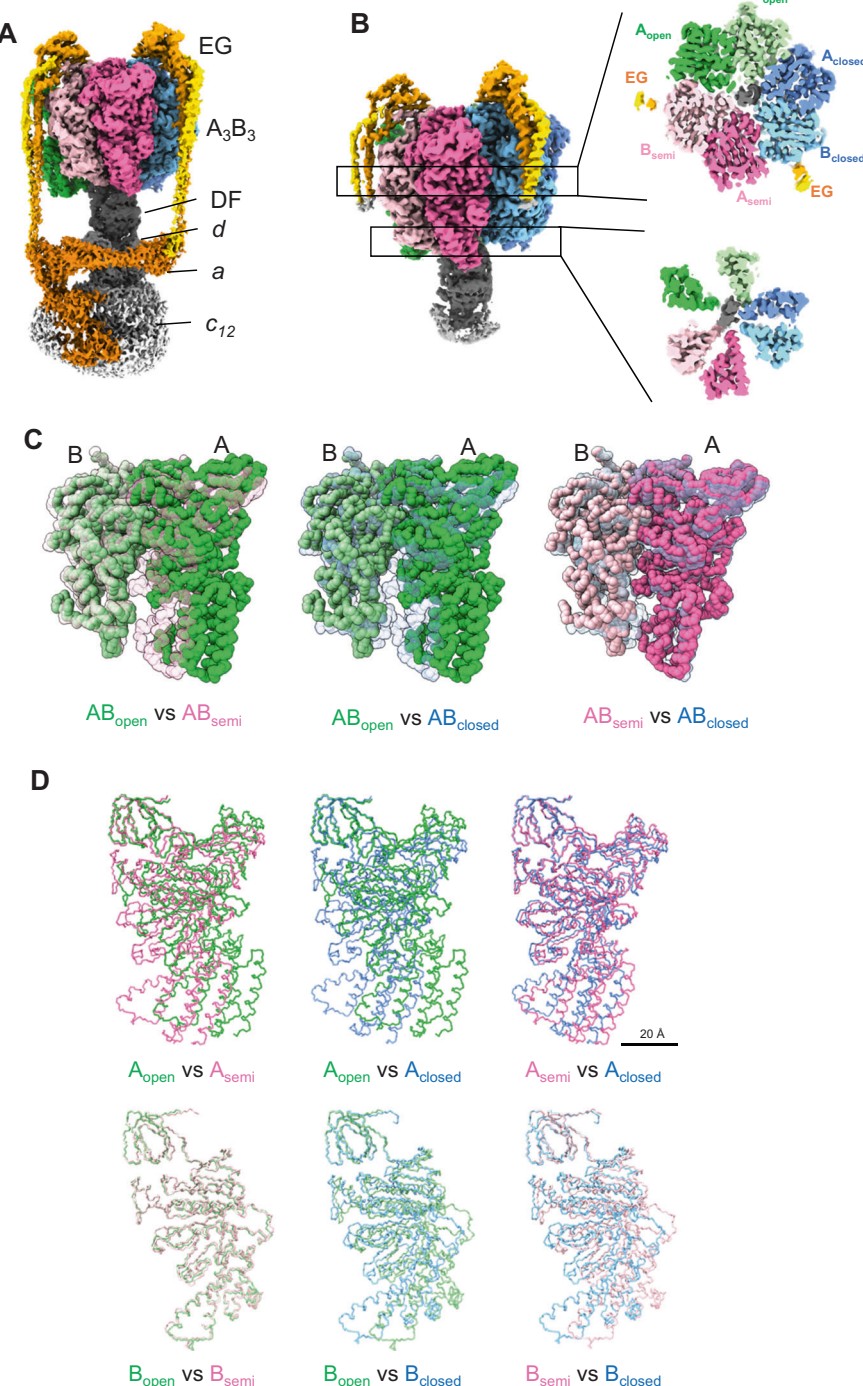

**Fig. 2 Cryo-EM density map and the atomic model for nucleotide-free V/A-ATPase. A** Cryo-EM density map of whole V/A-ATPase of state1 in the absence of nucleotide ($V_{nucfree}$ structure). **B** Cryo-EM density map of $V_1EG$ of state1 without nucleotide (*left*). Cross-sections of the nucleotide-binding sites (*right upper*) and $A_3B_3$ C-terminal region (*right lower*) viewed from the top. **C** Comparison of the AB dimer structures in $V_{nucfree}$. AB dimers are shown as space-filling models and superimposed on the β barrel domain (A subunit 1–70 a.a.). *Left;* $AB_{open}$ (solid) vs. $AB_{semi}$ (semi-transparent), *middle;* $AB_{open}$ (solid) vs. $AB_{closed}$ (semi-transparent), and $AB_{semi}$ (solid) vs. $AB_{closed}$ (semi-transparent). **D** Comparison between each AB subunit in $V_{nucfree}$. The subunits are shown as wire representations. A and B subunits are superimposed on the β barrel domain.

incubated for 120 s at 25 °C and then loaded onto a holey grid, followed by flash freezing.

We determined three rotational states followed by focused refinement using a $V_1EG$ mask for each state (Supplementary Fig. 3b). In the density maps obtained for each state, the amino acid residues of the nucleotide-binding sites in both $AB_{closed}$ and $AB_{semi}$ were well resolved, but the CHB domains of the AB

dimers were blurred due to structural heterogeneity, as with the $V_{nucfree}$. For state1, we identified an atomic resolution structure of state1-1 from 40,831 particles at 3.1 Å resolution and state1–2 from 28,801 particles at 3.2 Å resolution by further 3D classification without alignment (Supplementary Fig. 3b). The same classification analysis was performed for state2 and state3, yielding state 2–1 (3.0 Å resolution) and state 2–2 (3.4 Å

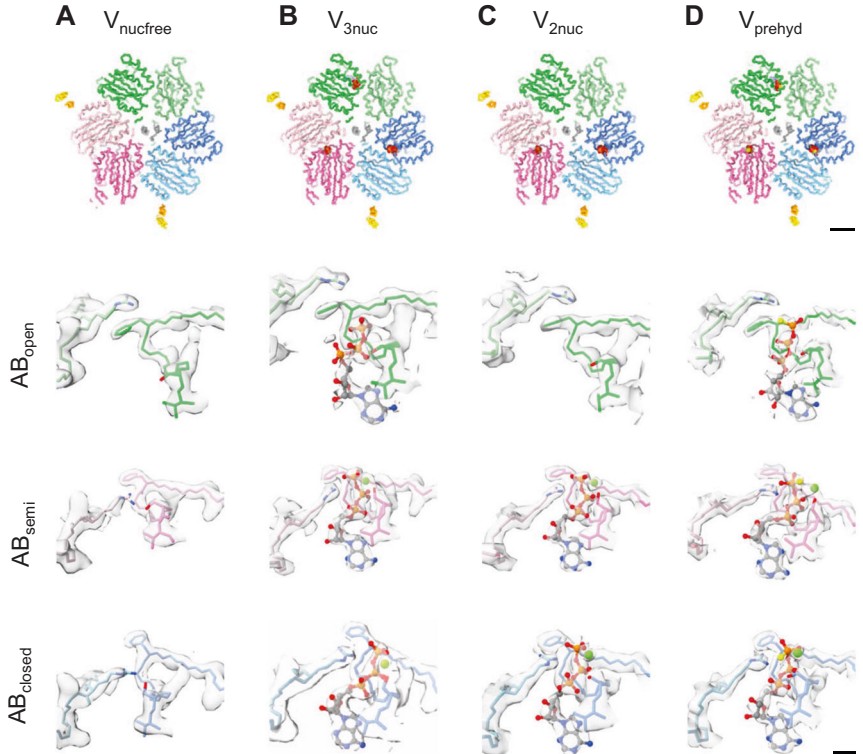

**Fig. 3 Structures of nucleotide-binding sites obtained in each condition.** *Upper panels:* $V_{nucfree}$ (**A**), $V_{prehyd}$ (**B**), $V_{3nuc}$ (**C**), and $V_{2nuc}$ (**D**) viewed from the cytosolic side. The scale bar is 20 Å. Magnified views of the three nucleotide-binding sites ($AB_{open}$, $AB_{semi}$, and $AB_{closed}$) in each structure are shown in the rows below. Cryo-EM maps are represented as semi-translucent. Bound nucleotides and Mg ions are shown in ball-and-stick and sphere representation, respectively. The scale bar is 4 Å.

resolution), and state 3–1 (3.0 Å resolution) and state 3–2 (3.4 Å resolution) respectively. In these structures, nucleotide densities have been identified in the three catalytic sites. Hereafter, we refer to the structures obtained at ATP saturating conditions as $V_{3nuc}$.

The structure of $AB_{open}$ of $V_{3nuc}$ state1–1 is almost identical to that of $V_{nucfree}$ state1–1 (Supplementary Fig. 6). This is confirmed by the fact that the *rmsd* values in the CHB of $A_{open}$ and $B_{open}$ for $V_{3nuc}$ state1–1 and $V_{nucfree}$ are less than 1 Å (Supplementary Tables 2 and 3). The $AB_{open}$ of state1–2 adopts a slightly more closed conformation compared to that of state1–1, which results from a movement of CHB of $B_{open}$ toward the β-barrel domain (Fig. 4C). Nevertheless, the $AB_{open}$ of $V_{3nuc}$ with bound ATP retains the interaction with the DF shaft, indicating that ATP binding to the $AB_{open}$ does not move the DF shaft.

The $AB_{semi}$ in state1–2 has a more closed structure than that in state1–1 mainly due to the movement of CHB in $B_{semi}$ (Fig. 4B, Supplementary Table 3). In summary, $V_{3nuc}$ state1–2 has a more closed structure than state1–1 due to the movement of the CHB of both $A_{semi}$ and $B_{semi}$, but the slightly closed conformation of state1–2 is independent of ATP binding to the $AB_{open}$.

**Structures of the catalytic sites at AB dimers of $V_{3nuc}$.** In both state1–1 and state1–2 structures obtained under ATP saturating conditions, a bound ATP molecule is clearly observed in the catalytic site of $AB_{open}$ (Fig. 3B and Supplementary Fig. 7c). The catalytic sites in the $AB_{closed}$ and $AB_{semi}$ in both the state1–1 and state1–2 also contained density corresponding to an ATP molecule, and in these cases, the associated magnesium ions were visible (Fig. 3B and Supplementary Fig. 7a, b). In the $V_{3nuc}$ structure, we did not find density corresponding to nucleotides between the D and A subunits as reported in a previous paper[25] (Supplementary Fig. 8).

In the catalytic site of $AB_{semi}$ of $V_{3nuc}$, the density of each nucleotide phosphate atom was easily identifiable (Supplementary Fig. 7b), indicating that the ATP molecule occupies the catalytic site in $AB_{semi}$. The protein structure is sufficiently clear to also provide a detailed picture of the configuration of the catalytic side chains (Fig. 5A–C). The γ-phosphate of ATP and the magnesium ion are coordinated by the A/K234 and A/S235 residues on the P-loop, which contains the conserved nucleotide-binding motif[26]. The aromatic ring of A/F230, not conserved in F type ATPase, is oriented away from the triphosphate moiety, allowing access of the guanidium group to the arginine finger (Supplementary Fig. 9). Considering clear EM density for the γ-phosphate of the ATP bound in $AB_{semi}$, hydrolysis of ATP is unlikely to proceed in $AB_{semi}$.

The nucleotide-binding site of the $AB_{closed}$ is shown in Fig. 5B. The geometry of ATP binding in $AB_{closed}$ is very similar to that found in the $AB_{semi}$, however, the carbonyl group of A/E257 is closer to the γ-phosphate by about 1 Å than $AB_{semi}$. In state1–2 of $V_{3nuc}$, the γ-phosphate of ATP bound to the $AB_{closed}$ appears to be separated from β-phosphate when a relatively high density threshold is used (Supplementary Fig. 10). These findings strongly suggest that ATP bound to $AB_{closed}$ is either already hydrolyzed or in the process of being hydrolyzed. The state of ATP in $AB_{closed}$ is discussed further in Supplementary Note.

In the nucleotide-binding site of the $AB_{open}$ of $V_{3nuc}$, the adenosine moiety of ATP is occluded as in the $AB_{semi}$ and $AB_{closed}$, with A/F415, A/Y500, and A/V236 forming the adenine binding pocket, however, the hydrogen bonding of the ribose moiety to the side chain of B/N363 is lost due to movement of the CHB of $A_{open}$ (Fig. 5C). Unlike in the $AB_{closed}$ and $AB_{semi}$, the phenyl group of A/230F in the $AB_{open}$ is closer to the triphosphate group of ATP due to the torsion of the main chain, resulting in the formation of a hydrophobic barrier between the

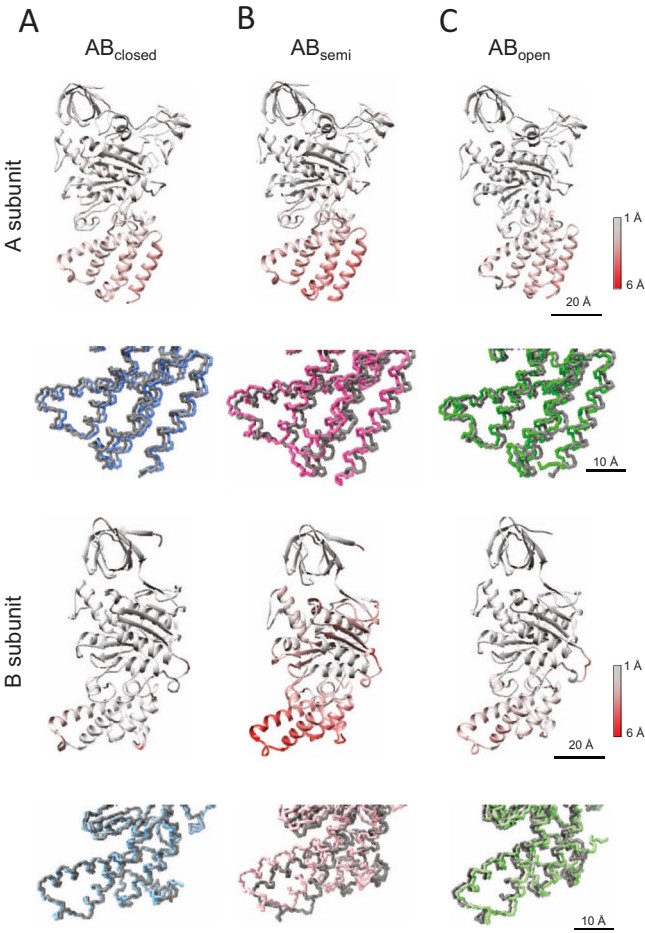

**Fig. 4 Comparison between state1–1 and 1–2 in each subunit of V$_{3nuc}$.** The subunits of state1–1 and 1–2 were superimposed on the β barrel domain (A: 1–70 a.a., B: 1–70 a.a.). Ribbon models are colored by the *rmsd* values calculated for the atoms of the main chain; gray (small changes) to red (large changes). Magnified views of the CHBs are represented in the lower panels as wire models. The models of state1–1 are represented in gray, and state1–2 are represented in different colors. **A** Subunits in AB$_{closed}$, **B** subunits in AB$_{semi}$, **C** subunits in AB$_{open}$.

catalytic side chains and the triphosphate moiety of ATP (Supplementary Fig. 9). Compared with AB$_{semi}$, the side chains of catalytic residues A/E257, A/R258, and B/R360 of AB$_{open}$ are much further away from the γ-phosphate of ATP (10.0, 10.0, and 7.1 Å, respectively). Consequently, the configuration of the catalytic residues in the nucleotide-binding site of AB$_{open}$ is not appropriate for hydrolysis of the bound ATP. Instead, the bound ATP has the potential to zipper the AB interface via interaction with the surrounding catalytic residues, which ultimately results in the transition of AB$_{open}$ to a more closed form via a typical zipper conformational change.

**Structures of V/A-ATPase waiting for ATP to bind.** To determine the ATP-waiting structure of V/A-ATPase, we prepared a cryo-grid with a reaction mixture containing 4 μM enzyme, 50 μM ATP, and the ATP regeneration system, pre-incubated for 300 s. We reconstructed three rotational states (state1, 2.7 Å, state2, 3.3 Å, and state3 3.6 Å resolution) from the single-particle images of the holo-complex and three rotational states of V$_1$EG (state1, 2.8 Å, state2, 3.1 Å, and state3, 2.8 Å) by focused masked refinement. For state1, two substates (state1–1, 2.9 Å and state1–2, 3.0 Å) were separated by focused classification using AB$_{open}$ and AB$_{semi}$ masks (Supplementary Fig. 3c).

In both AB$_{semi}$ and AB$_{closed}$ of state1–1, the apparent density of ATP-magnesium was observed, but the density of the γ-phosphate at the AB$_{closed}$ is weaker than that at the AB$_{semi}$ (Fig. 3C and Supplementary Fig. 7a, b). In contrast, density was not observed in the nucleotide-binding site of AB$_{open}$. For state1–2, as in state1–1, nucleotides are present in both AB$_{semi}$ and AB$_{closed}$, while AB$_{open}$ is empty. Hereafter we refer to the structure as V$_{2nuc}$.

The overall structure and geometry of the catalytic residues of state1–1 of V$_{2nuc}$ are largely identical to state1–1 of V$_{nucfree}$ and V$_{3nuc}$ (Supplementary Fig. 11). This structural similarity between V$_{2nuc}$, V$_{nucfree}$, and V$_{3nuc}$ is confirmed by the low *rmsd* values when comparing the CHB of the A and B subunits of these structures (Supplementary Tables 2 and 3). The similarity of the structures of these substrates indicates that the structural polymorphism of the V$_1$ domain is independent of the binding of ATP to AB dimers.

**Structures obtained at a saturating concentration of ATPγS (V$_{prehyd}$).** The V$_1$-ATPase from *T. thermophilus* is capable of hydrolyzing ATPγS, however, the turnover rate of ATPγS is much lower than that of ATP due to the decrease in hydrolysis rate[18] (Supplementary Fig. 2d). Thus, pre-hydrolysis structures of V/A-ATPase can be obtained at 4 mM ATPγS. The cryo-grid was prepared by blotting of the reaction mixture comprising Nucleotide-free V/A-ATPase and 4 mM of ATPγS in the absence of the regenerating system in order to exclude any effect of regenerated ATP produced from hydrolyzed ATPγS. We reconstructed three rotational states from the acquired EM images using the CRYOARM300 (JEOL). After the focused masked refinement of the V$_1$EG domain, we obtained atomic resolution structures of each state (state1, 2.7 Å, state2, 3.4 Å, and state3, 3.6 Å), respectively (Supplementary Fig. 3d). We refer to these structures as V$_{prehyd}$. For state1, two sub-states, state1–1 and state1–2, were obtained at 2.7 and 2.9 Å resolution respectively, by focused classification using a mask with AB$_{open}$ and AB$_{semi}$ (Supplementary Fig. 3d). For the AB$_{open}$ of V$_{prehyd}$, bound ATPγS is clearly observed in the catalytic site, which has an almost identical structure to that of the ATP bound state of V$_{3nuc}$ (Fig. 3B).

The nucleotide-binding sites of the AB$_{closed}$ and AB$_{semi}$ of V$_{prehyd}$ are almost identical to those of V$_{3nuc}$, respectively, as shown in Fig. 5. The γ-phosphate group of the bound ATPγS molecule in the AB$_{semi}$ is well resolved as seen in for ATP in V$_{3nuc}$ and V$_{2nuc}$ (Fig. 3d and Supplementary Fig. 7b). In contrast, the density of γ-phosphate of ATPγS at the AB$_{closed}$ is faint (Supplementary Fig. 7a), suggesting that the ATPγS in the AB$_{closed}$ has already been hydrolyzed and the bound nucleotide in the AB$_{closed}$ is ADP. This indicates that the AB$_{semi}$ is in the pre-hydrolysis conformation, waiting for ATP hydrolysis.

**Discussion**

We have obtained catalytic intermediates of the V$_1$ domain, V$_{nucfree}$, V$_{3nuc}$, V$_{2nuc}$, and V$_{prehyd}$, with different nucleotide occupancy. Despite the different nucleotide occupancy of these structures, their overall conformations are very similar. For instance, the *rmsd* of the C$_α$ chains of A$_3$B$_3$ in V$_{nucfree}$ and V$_{3nuc}$ state1–1 is 1.98 Å. In addition, the relative position of the central DF shaft within the asymmetric A$_3$B$_3$ is almost the same in the V$_{nucfree}$ and V$_{3nuc}$ structures. These findings demonstrate that the configuration between the DF shaft and individual AB dimers is independent of the state of nucleotide occupancy of each AB dimer. In other words, the structure of the V$_1$ domain adopts three rotational states, 1–3, during continuous ATP hydrolysis, with the conformational changes in the A$_3$B$_3$ hexamer driven by ATP hydrolysis, being discrete rather than continuous.

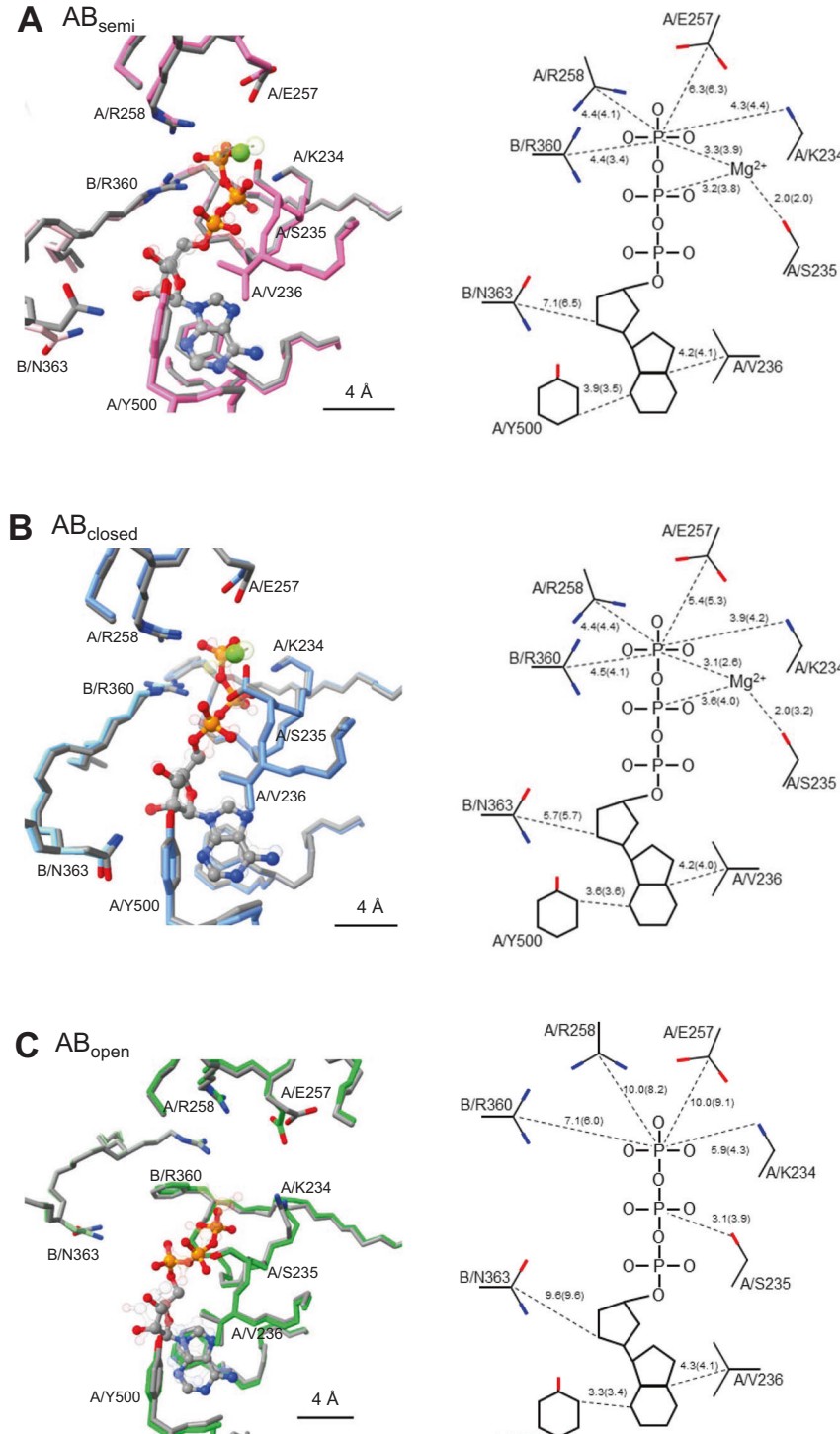

**Fig. 5 Coordination of nucleotides in the binding sites of $V_{3nuc}$ and $V_{prehyd}$.** *Left panels;* Comparison of the three nucleotide-binding sites ($AB_{open}$ (**A**), $AB_{semi}$ (**B**), and $AB_{closed}$ (**C**)) of state1–1 of $V_{3nuc}$ shown with colored (green, blue, and pink) atoms and bonds, and main chain, and state1–1 of $V_{prehyd}$ shown with gray atoms, bonds and main chain. *Right panels;* schematic representations of the coordination of the ATP group in the three binding sites of $V_{3nuc}$ and $V_{prehyd}$ in parentheses. The distances between the atoms are shown in dotted lines. All distances are in Å.

The $V_{3nuc}$ structure, obtained under ATP saturation conditions shows all three catalytic sites occupied by ATP or the products of hydrolysis (ADP + $Pi$). Since the hydrolyzed $Pi$ is clearly visible in the $AB_{closed}$ of $V_{3nuc}$ (Supplementary Fig. 10), it is assumed that $V_{3nuc}$ is the structure before dissociation of $Pi$ from the catalytic site in $AB_{closed}$. We also obtained the $V_{2nuc}$ structure in which ATP and product(s) are bound to $AB_{semi}$ and $AB_{closed}$, respectively, but $AB_{open}$ is empty. The $V_{2nuc}$ is therefore assumed to be the

structure of the protein awaiting ATP binding to $AB_{open}$. When using ATPγS as a substrate, which has a very slow hydrolysis rate, the high-resolution atomic structure of the $V_1$ domain allowed visualization of ATPγS molecules bound to the catalytic sites of $AB_{open}$ and $AB_{semi}$, as well as identification of the hydrolyzed ATPγS at the $AB_{closed}$. The $V_{prehyd}$ reveals both that the $AB_{closed}$ adopts the post-hydrolysis state where the product of phosphate ($Pi$) is dissociated, and that the $AB_{semi}$ is awaiting ATP hydrolysis.

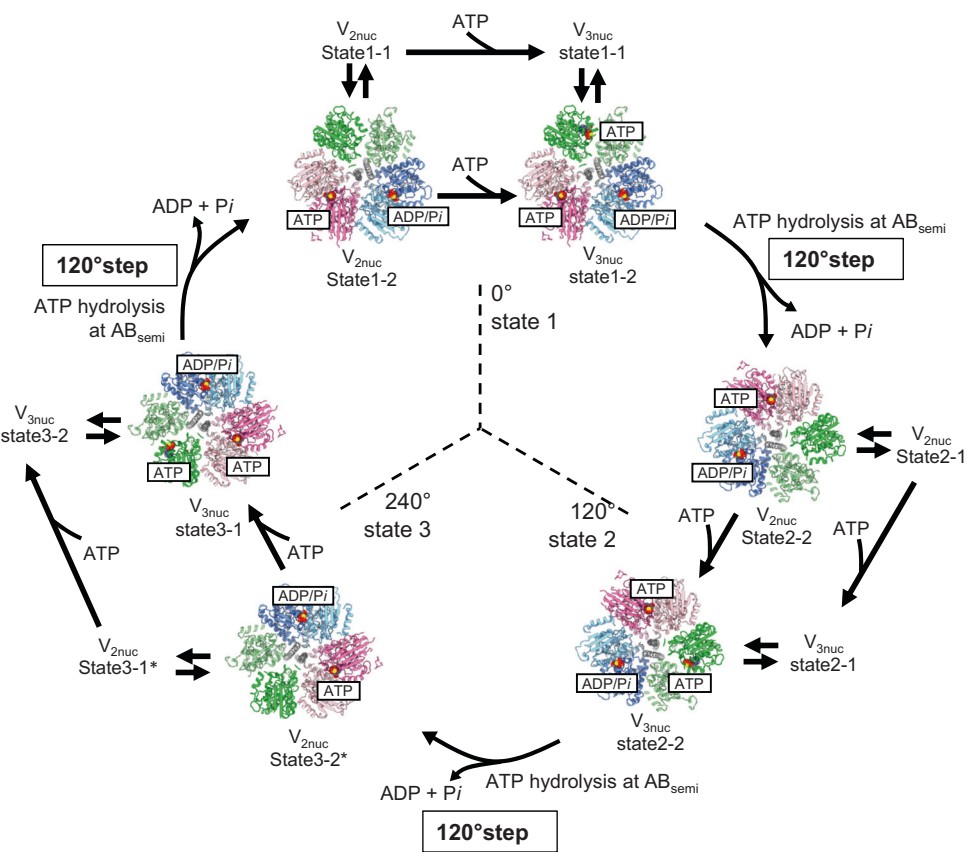

**Fig. 6 Chemo-Mechanical cycle of V/A-ATPase driven by ATP hydrolysis.** The structures of V/A-ATPase viewed from the cytosolic side are shown as ribbon models. The coiled-coil of the DF subunits is shown in gray. The bound ATP molecules are highlighted in sphere representations. State1–1 and 1–2 of $V_{2nuc}$ are in equilibrium and are fluctuating. These structures transit to state1–1 and 1–2 of $V_{3nuc}$ by ATP binding to $AB_{open}$, without a 120° rotation step of the DF rotor. $V_{3nuc}$ in state1–1 and state1–2 are also in equilibrium. ATP hydrolysis at $AB_{semi}$ and zipper motion at $AB_{open}$ occur simultaneously. This triggers the transition of $V_{3nuc}$ in state1–2 to $V_{2nuc}$ state2–2 together with the 120° rotation step and simultaneous release of ADP and $Pi$. State2 of $V_{2nuc}$ returns to state1 via state3 of $V_{2nuc}$ by the same process. Asterisks indicate the structures which were not identified in this study.

The structures provide important insights into the chemo-mechanical cycle of V/A-ATPase. The V/A-ATPase undergoes a unidirectional conformational change from state1 to state2 to state3 when powered by ATP. Thus, $V_{3nuc}$ of state1, in which three catalytic sites are already occupied by nucleotides, should change to state2 of $V_{2nuc}$, following ATP hydrolysis at $AB_{semi}$, and the subsequent or simultaneously dissociation of ADP and $Pi$ by the discrete structural transition of $AB_{closed}$ to $AB_{open}$ (Fig. 6). This demonstrates that rotation of the rotary ATPase proceeds via the tri-site model with the protein progressing through a two nucleotide bound state and a three-nucleotide bound state, settling the long-standing debate on whether the bi-site model or tri-site model is appropriate for rotary ATPases[5,6,27–30].

Based on previous single molecular observation experiments for both $F_1$- and $V_1$-ATPase, ATP binding onto the enzyme directly triggers the first 120° rotation step of the DF shaft[18,19,30] (Supplementary Fig. 1a). In light of our findings presented here, this scheme needs to be redrawn; the rotor does not immediately travel 120° as a result of ATP binding to enzyme.

The next catalytic event after ATP binding is ATP hydrolysis in $AB_{semi}$. Each conformational change, from $AB_{open}$ to $AB_{semi}$, $AB_{semi}$ to $AB_{closed}$, and $AB_{closed}$ to $AB_{open}$ occurs simultaneously, with the rotation of the shaft, and with the hydrolysis of ATP in the $AB_{semi}$ and release of products (ADP and $Pi$) from the $AB_{closed}$ (Fig. 6 and Supplementary Movie 1). This is in contrast to the classical rotary model, where catalytic events occur in sequence at the three catalytic sites, until now the broadly accepted mechanism of action of the $F_1$-ATPase[6,27,31,32].

In the $V_{2nuc}$ and $V_{3nuc}$, two sub-states, state1–1 and state1–2 were identified (Supplementary Fig. 12). These substrates were also identified in $V_{nucfree}$, thus the conformational dynamics of the $V_1$ domain are independent of ATP binding. In other words, state1–1 and state1–2 are in a thermal equilibrium state, irrespective of nucleotide occupancy in each catalytic site. Both $AB_{semi}$ and $AB_{open}$ in state1–2 adopt more closed structures than those in state1–1, suggesting that state1–2 of $V_{3nuc}$ is likely an intermediate structure just prior to the 120° rotation step of the DF shaft. Compared to state1–2 of $V_{3nuc}$, state1–2 of $V_{prehyd}$ exhibits slightly more closed structures of $AB_{open}$ and $AB_{semi}$ (Supplementary Fig. 13), likely to be associated with the progress of the catalytic reaction in $AB_{closed}$, i.e., the dissociation of the phosphate. In this respect, state1–2 of $V_{prehyd}$ may be another reaction intermediate structure in which the $Pi$ in the $AB_{closed}$ is released prior to the 120° step (Supplementary Fig. 14).

Based on the catalytic intermediates of the $V_1$ domain of V/A-ATPases obtained under four different reaction conditions, we propose a model for the ATP-driven rotation mechanism of V/A-ATPases (Fig. 7, Supplementary Fig. 15).

When ATP binds to $V_{nucfree}$, which is in a stable initial state (ground state), the enzyme transits into the steady-state for ATP hydrolysis (Fig. 7, *upper row*). The $V_{3nuc}$ structure is formed by binding of ATP to the $AB_{open}$ of $V_{2nuc}$ but the binding of ATP itself does not cause structural transitions between AB dimers associated with the 120° step of the DF shaft.

In the $V_{3nuc}$ structure, where nucleotides are bound to all three AB dimers, three distinct but associated catalytic events occur at

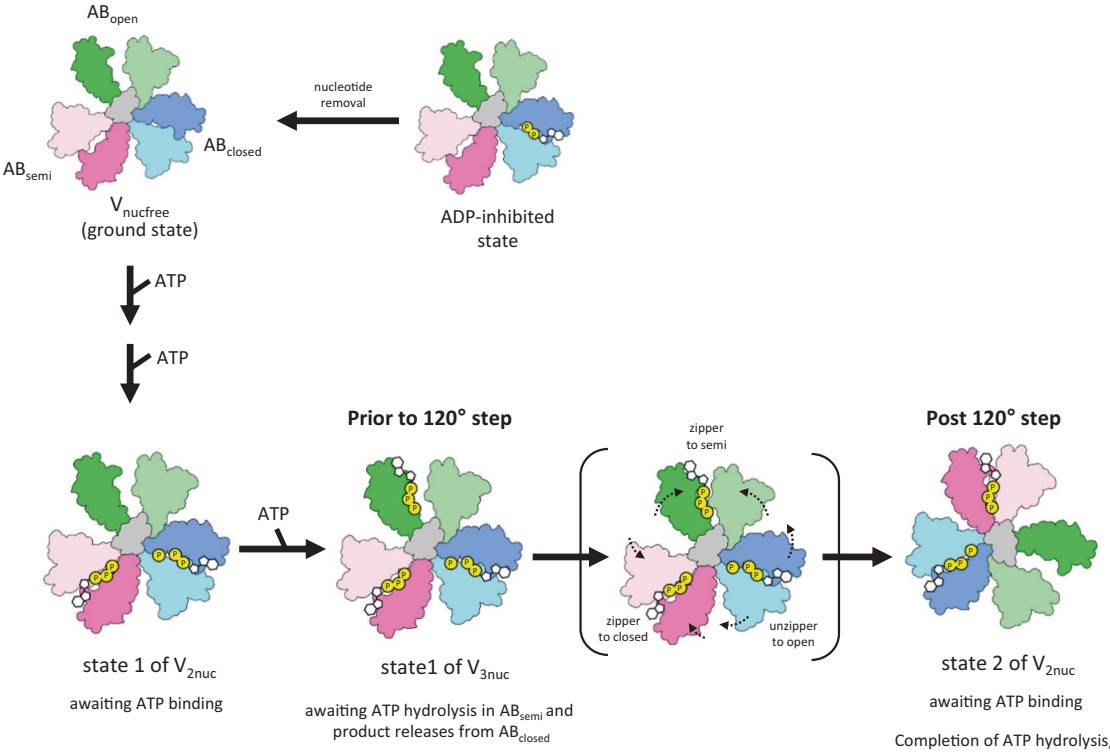

**Fig. 7 The rotary mechanism of V/A-ATPase powered by ATP hydrolysis.** The schematic models of $AB_{open}$, $AB_{semi}$, and $AB_{closed}$ are shown in green, pink, and blue, respectively. The coiled-coil region of the D subunit in contact with $A_3B_3$ is shown in gray. In the ADP inhibited state, the entrapped ADP in $AB_{closed}$ hampers the structural transition of $AB_{open}$ to $AB_{semi}$ by binding of ATP to $AB_{open}$. The $V_{nucfree}$ in the ground state is activated by the binding of ATP to the catalytic sites. In $V_{2nuc}$ awaiting ATP binding, binding of ATP to $AB_{open}$ does not induce the 120° rotation step. In $V_{3nuc}$, both zipper motion of $AB_{open}$ and ATP hydrolysis in $AB_{semi}$ induce unzipper motion of $AB_{closed}$ accompanying the release of ADP and $Pi$. The catalytic events in the three AB dimers occur simultaneously with the 120° step of the DF shaft, resulting in the structural transition of state1 of $V_{3nuc}$ to state2 of $V_{2nuc}$.

the three AB dimers simultaneously and these events are coupled to the first 120° rotation step of the DF shaft. One of the driving events for this transition is the conformational change from the ATP-bound $AB_{open}$ to the more closed $AB_{semi}$, which can be explained by a zipper motion of $AB_{open}$ occurring upon ATP binding. From our structures, a comparison between $AB_{open}$ and $AB_{semi}$ implies that new hydrogen bonds form between the triphosphate moiety of ATP and the surrounding side chain groups of B/R360, A/258R, A/257E, and A/K234 (Fig. 5).

The $V_{prehyd}$ structures indicate that the ATP bound to $AB_{semi}$ is awaiting hydrolysis. The conformational change from $AB_{semi}$ to $AB_{closed}$ should occur spontaneously because it involves ATP hydrolysis, an exergonic reaction. In contrast, ADP bound to $AB_{closed}$ hampers the unzipper motion in $AB_{closed}$, thereby preventing the overall structural transition of the $V_1$ domain. This is supported by the fact that V/A-ATPase adopts the ADP inhibited state in which ADP is entrapped in $AB_{closed}$ (Fig. 7, upper line). The enzyme in the ADP-inhibited state does not show ATP hydrolysis activity even at saturated ATP concentration[13,16].

In summary, the ATP-driven unidirectional rotation of V/A-ATPase proceeds by a discrete structural transition between the three rotational states, i.e., the potential barrier to the structural transition of $AB_{closed}$ to $AB_{open}$, accompanied by the release of ADP and $Pi$, is overcome by both a zipper motion of $AB_{open}$ by the bound ATP and ATP hydrolysis in $AB_{semi}$. Since the ATP hydrolysis reaction is a heat dissipation process, the structural transition of $AB_{semi}$ to $AB_{closed}$ associated with the ATP hydrolysis occurs spontaneously and irreversibly, resulting in a unidirectionality of the 120° steps of the rotor. In other words, our model explains the unidirectional rotation by a ratchet-like

mechanism driven by ATP hydrolysis, rather than the power stroke model proposed previously for $F_1$-ATPase[5,33].

V/A-ATPase and $F_oF_1$ are molecular machines based on the same construction principle, and thus are likely to share the same rotary mechanism. In fact, observation of rotation with high time-resolved rotation analysis using a tiny gold rod showed that bacterial F-ATPase and an A-ATPase share the same rotation mechanism[34]. For both rotary ATPases, a 120° rotation step together with ATP hydrolysis occurs after the catalytic dwell under ATP-saturated conditions. Importantly, for the thermophilic $F_1$, the first 120° rotation step also includes ~80° and ~40° substeps, suggesting the existence of at least one additional catalytic intermediate of $F_1$[5,27,35]. A recent structural study of thermophilic $F_1$-ATPase indicated a possible intermediate structure responsible for the substeps[32]. In the V/A-ATPase, any intermediate structure containing phosphate after ADP or $Pi$ release is likely to be in an unstable state, and therefore studies on the ATP-driven rotation of V/A-ATPase have failed to reveal the presence of any substep[19].

## Methods

**Preparation of *Tth* V/A-ATPase for biochemical assay and cryo-EM imaging.** The *Tth* V/A-ATPase containing His3 tags on the C-terminus of each c subunit and the TSSA mutation (S232A and T235S) on the A subunit was isolated from *T. thermophilus* membranes as previously described[24] with the following modifications. The enzyme, solubilized from the membranes with 10% Triton X-100 was purified by Ni$^{2+}$-NTA affinity with 0.03% dodecyl-β-D-maltoside (DDM). For bound nucleotide removal, the eluted fractions containing *Tth* V/A-ATPase were dialyzed against 200 mM Sodium phosphate, pH 8.0, 10 mM EDTA, and 0.03% DDM overnight at 25 °C with three buffer changes, followed by dialysis against 20 mM Tris-Cl, pH 8.0, 1 mM EDTA, and 0.03% DDM (TE buffer) prior to anion exchange chromatography using a 6 ml Resource Q column (GE healthcare). The *Tth* V/A-ATPase was eluted by a linear NaCl gradient using a TE buffer

(0–500 mM NaCl, 0.03% DDM). The eluted fractions containing *holo*-*Tth* V/A-ATPase were concentrated to ~10 mg/ml using Amicon 100k molecular weight cut-off filters (Millipore). For nanodisc incorporation, the 1,2-Dimyristoyl-sn-glycero-3-phosphorylcholine (DMPC, Avanti) was used to form lipid bilayers in reconstruction as previously described[16]. Purified *Tth* V/A-ATPase solubilized in 0.03% n-DDM was mixed with the lipid stock and membrane scaffold protein MSP1E3D1 (Sigma) at a specific molar ratio $V_OV_1$:MSP:DMPC lipid = 1:4:520 and incubated on ice for 0.5 h. Then, 200 μL of Bio Beads SM-2 equilibrated with a wash buffer (20 mM Tris-HCl, pH8.0, 150 mM NaCl) was added to the 500 μL mixture. After 2 h incubation at 4 °C with gentle stirring, an additional 300 μL of Bio Beads was added and the mixture was incubated overnight at 4 °C to form the nanodiscs. The supernatant of the mixture containing nanodisc-*Tth* V/A-ATPase (nd-V/A-ATPase) was loaded onto the Superdex 200 Increase 10/300 column equilibrated with wash buffer. The peak fractions were collected, analyzed by SDS-PAGE, and concentrated to ~4 mg/mL. The prepared *nd*-V/A-ATPase was immediately used for biochemical assay or cryo-grid preparation since *nd*-V/A-ATPase aggregates within a few days.

**Biochemical assay**. The quantitative analysis of bound nucleotides of *Tth* V/A-ATPase was carried out using anion-exchange high-performance liquid chromatography[13]. Bound nucleotides were released from the enzyme by the addition of 5 μl of 60% perchloric acid to 50 μl of the enzyme solution. Thereafter, the mixture was incubated on ice for 10 min. Then, 5 μl of 5 M $K_2CO_3$ solution was added and the mixture was incubated on ice for 10 min. The resulting pellet was removed by centrifugation at 4 °C. The supernatant was applied to a Cosmopak-200 column equilibrated with 0.1 M sodium phosphate buffer (pH 7.0). The column was eluted isocratically with the same buffer at a flow rate of 0.8 ml/min. The nucleotide was monitored at 258 nm. The peak area was determined by automatic integration.

ATPase activity was measured at 25 °C with an enzyme-coupled ATP-regenerating system, as described previously[13]. The reaction mixture contained 50 mM Tris-HCl (pH 8.0), 100 mM KCl, different concentrations of ATP-Mg, 2.5 mM phosphoenolpyruvate (PEP), 50 μg/ml pyruvate kinase (PK), 50 μg/ml lactate dehydrogenase, and 0.2 mM NADH in a final volume of 2 ml. The reaction was started by the addition of 20 pmol *nd* V/A-ATPase to 2 ml of the assay mixture, and the rate of ATP hydrolysis was monitored as the rate of oxidation of NADH was determined by the absorbance decrease at 340 nm.

**Cryo-EM imaging of *Tth* V/A-ATPase**. Sample vitrification was performed using a semi-automated vitrification device (Vitrobot, FEI). For *nd*-V/A-ATPase that underwent nucleotide removal, hereafter referred to as nucfree *nd*-V/A-ATPase, 2.4 μl of sample solution at a concentration of 3 mg/ml (2 μM) was applied to glow discharged Quantifoil R1.2/1.3 molybdenum grid discharged by Ion Bombarder (Vacuum Device) for 1 min. The grid was then automatically blotted once from both sides with filter paper for 6 s blot time. The grid was then plunged into liquid ethane with no delay time.

The reaction basal buffer (RB buffer) containing 50 mM Tris-Cl, pH 8.0, 100 mM KCl, and 2 mM $MgCl_2$ was used for different reaction conditions. For saturated ATP or ATP waiting condition, 4 μM of nucfree *nd*-V/A-ATPase was mixed with the same volume of ×2 RB buffer containing 10 mM PEP, 200 μg/ml of PK, 12 mM, or 100 μM of ATP-Mg. Then the mixtures were incubated for 120 or 300 s at 25 °C, followed by blotting and vitrification, respectively. For the ATPγS saturated condition, 4 μM of nucfree *nd*-V/A-ATPase was mixed with the same volume of ×2 RB buffer containing 8 mM ATPγS-Mg, then incubated for 300 s at 25 °C, followed by the blotting and vitrification.

With the exception of the saturated ATPγS condition, cryo-EM imaging was performed with a Titan Krios (FEI/Thermo Fisher) operating at 300 kV acceleration voltage and equipped with a direct K3 (Gatan) electron detector in electron counting mode (CDS). Data collection were carried out using SerialEM software[36] at a calibrated magnification of 0.88 Å pixel$^{-1}$ (×81,000) and a total dose of 50.0 e$^-$ Å$^{-2}$ (or 1.0 e$^-$ Å$^{-2}$ per frame) (where e$^-$ specifies electrons) with a total 5 s exposure time. The defocus range was −0.8 to −2.0 μm. The data were collected as 50 movie frames.

For the saturated ATPγS condition, Cryo-EM movie collection was performed with a CRYOARM 300 (JEOL) operating at 300 keV accelerating voltage and equipped with a K3 (Gatan) direct electron detector, in electron counting mode (CDS) using the data collection software serialEM. The pixel size was 1.1 Å/pix (×60,000) and a total dose of 50.0 e$^-$ Å$^{-2}$ (1.0 e$^-$ Å$^{-2}$ per frame) with a total 3.0 s exposure time (50 frames) with a defocus range of −1.0 to −3.5 μm.

**Image processing**. Image processing steps for each reaction condition are summarized in Supplementary Fig. 3a–d. Image analysis software, Relion 3.1 and Cryosparc 3.2, were used[37,38]. CTFFIND 4.1 and MotionCor2 were used for CTF estimation and movie correction in Relion[39,40]. Topaz software was used for machine-learning-based particle picking[41]. We started with 15,317 movies for the nucleotide-free enzyme (nucfree *nd*-V/A-ATPase), 13,164 movies for saturated ATP condition, 15,711 movies for saturated ATP condition, and 17,522 movies for ATPγS condition. The software used in the steps is indicated in the figure. Autopicking based on template matching or based on Topaz machine-learning resulted in 4,354,341 particles for the nucfree *nd*-V/A-ATPase, 2,300,834 particles for the ATP saturated condition, 1,671,397 particles for the ATP waiting condition,

and 4,677.284 particles for the ATPγS waiting condition. Particles were extracted at 5x the physical pixel size from the movie-corrected micrographs and selected using 2D or 3D classification (nucfree nd-V/A-ATPase; 132,904 particles, saturated ATP; 188,673 particles, ATP waiting; 186,928 particles, ATPγS; 197,960 particles). The selected particles were extracted at full pixel size and subjected to 3D auto-refinement refollowed by CTF refinement by Bayesian polishing. Another round of 3D auto-refine, CTF refinement, and a final round of masked auto-refinement gave *holo*-V/A-ATPase maps at between 2.7 and 6.3 Å resolution. The membrane domain was visible but not particularly clear compared to the hydrophilic $V_1$ domain in a *holo*-enzyme map. This seemed to be due to the structural flexibility between the membrane domain and $V_1$ in the *holo*-enzyme. Focused refinement with signal subtraction targeting the $V_1EG$ region improved the map quality of the $V_1EG$ region (Supplementary Fig. 3a–d). The refinements provided the density maps for $V_1EG$ under each condition at 2.8–4.1 Å resolution. After the focused refinement, masked classification on $A_{open}$ and $B_{semi}$ subunits was carried out to classify the conformational differences. The resolution was based on the gold standard Fourier shell correlation = 0.142 criterion.

**Model building and refinement**. To generate the atomic model for the $V_1EG$ region of V/A-ATPase, the individual subunits of the $V_1EG$ model from the previous structure of V/A-ATPase (PDBID: 6QUM) were fitted into the density map as rigid bodies[25] with particular focus on the N terminal region of EG stalk (E; 1–77 aa., G; 2–33 aa). The rough initial model was refined against the map with Phenix suite phenix.real_space_refine program[42]. The initial model was extensively manually corrected residue by residue in COOT[42] in terms of side-chain conformations. Peripheral stalks were removed due to low resolution in this region. The corrected model was again refined by the phenix.real_space_refine program with secondary structure and Ramachandran restraints, then the resulting model was manually checked by COOT. This iterative process was performed for several rounds to correct remaining errors until the model was in good agreement with geometry, as reflected by the MolProbity score of 1.08–1.74 and EMRinger score of 1.59–3.94[43,44]. For model validation against over-fitting, the built models were used for calculation of FSC curves against both half maps, and compared with the FSC of the final model against the final density map used for model building by phenix.refine program. The statistics of the obtained maps and the atomic model were summarized in Supplementary Tables 4–11. RMSD values between the atomic models were calculated using UCSF chimera[45]. All the figures were rendered using UCSF chimeraX[46].

**Reporting summary**. Further information on research design is available in the Nature Research Reporting Summary linked to this article.

## Data availability

The cryo-EM maps have been deposited in the EMDB under accession codes 31841, 31842, 31843, 31844, 31845, 31846, 31847, 31848, 31849, 31850, 31851, 31852, 31853, 31854, 31855, 31856, 31857, 31858, 31859, 31860, 31861, 31862, 31863, 31864, 31865, 31866, 31867, 31868, 31869, 31870, 31871, 31872, and 31873. The atomic models have been deposited in the Protein Data Bank under accession codes 7VAI, 7VAJ, 7VAK, 7VAL, 7VAM, 7VAN, 7VAO, 7VAP, 7VAQ, 7VAR, 7VAS, 7VAT, 7VAU, 7VAV, 7VAW, 7VAX, 7VAY, and 7VB0. The initial model for model building is accessible in PDB under accession number 6QUM. The data that support the findings of this study are available from the corresponding author upon reasonable request.

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

## Acknowledgements
We are grateful to all the members of the Yokoyama Lab for their continuous support and technical assistance. Our research was supported by Grant-in-Aid for Scientific Research (JSPS KAKENHI) Grant Number 20H03231 to K.Y., 20K06514 to J.K., and Grant-in-Aid for JSPS Fellows Grant No. 20J00162 to A. Nakanishi, and Takeda Science Foundation to K.Y. Our research was also supported by Platform Project for Supporting Drug Discovery and Life Science Research (Basis for Supporting Innovative Drug Discovery and Life Science Research (BINDS)) from AMED under Grant No. JP17am0101001 (Support No. 1312), Grants-in-Aid from "Nanotechnology Platform" of the Ministry of Education, Culture, Sports, Science and Technology (MEXT) to K.M. (Project No. 12024046), and the Research Program for Next Generation Young Scientists of "Five-star Alliance" in "NJRC Mater. & Dev." under Grant No. 20215008 to A. Nakano.

## Author contributions
K.Y., J.K., A. Nakanishi, and A. Nakano designed, performed, and analyzed the experiments. J.K., A. Nakanishi, K.Y., A. Nakano, A.F., and S.S. analyzed the data and contributed to the preparation of the samples. T.K. and K.M. provided technical support and conceptual advice. K.Y. designed and supervised the experiments and wrote the paper. All authors discussed the results and commented on the paper.

## Competing interests
The authors declare no competing interests.
