## [Peer Review File · Nature Communications]

Structural snapshots of V/A-ATPase reveal the rotary catalytic mechanism of rotary ATPasesREVIEWER COMMENTS

Reviewer #1 (Remarks to the Author):

The manuscript by Kishikawa et al., provides significant new structural details on the V/A-ATPase and is an impressive study in terms of the data collected and the number of structures determined. This work will be significant in the field and should provide new insights into the V/A-ATPase and its comparison to other members of the field. With advances in cryoEM has come a depth of new information on rotary ATPases and their mechanism and this work will contribute to those studies. There are a few areas which it would be good to see considered and discussed within the manuscript;

How relevant are the concentrations of nucleotides used compared to what would be expected in the cell and how might differences influence the structure.

It is good to see a nanodisc being used to recreate a more native like membrane bilayer. Why is Vo more poorly resolved? Although a lower resolution why is the analysis of the different states in V1 not carried through into Vo. Are there any interesting observations in Vo which reflect the different nucleotide bound states?

It would be good to see a broader analysis for what has been shown with the F and V ATPase family to expand on the impact.

Line 121, states 1 and 2 were improved, what about state 3?

Line 151 although high resolution, ~3Å is not typically referred to as "atomic" resolution.

Line 288, how likely is it that this system has 3 ATP's bound at any one time, how often would it naturally experience "saturation" concentrations?

Line 306 demonstrated not demonstrate

Line 648, space after Cryo-EM

Reviewer #2 (Remarks to the Author):

Title: Structural snapshots of V/A-ATPase reveal a new paradigm for rotary catalysis

Journal: Nature Communications, NCOMMS-21-42328-T

All Authors: J. Kishikawa, A. Nakanishi, A. Nakano, S. Saeki, A. Furuta, T. Kato

K. Mistuoka, and K. Yokoyama

The primary source of ATP in archaea or the thermophilic bacterium, *Thermus thermophilus* (Tth) is the A-ATP synthase (in the manuscript called V/A-ATPase), which shares several structural features with eukaryotic V-ATPases, and is evolutionarily distant from the F-ATP synthases that fulfill this role in other bacteria and eukaryotes. Metabolism in archaea is coupled to the generation of a H⁺- and/or Na⁺- potential across the membrane, both of which can provide the energy for ATP synthesis by the A-ATP synthase. Whereas the F-ATP synthases in prokaryotes and eukaryotes catalyze ATP synthesis at the expense of an electrochemical ion potential, the evolutionarily related V-ATPases function as ATP-driven ion pumps, and are unable to synthesize ATP under physiological condition. In recent years, the Yokoyama lab has contributed significantly to the structural and mechanistic understanding of the Tth V/A-ATPase.

In the study presented by Kishikawa et al., the authors describe the atomic cryo-EM structures of four different reaction conditions using the mutant enzyme A/S232A, T235S, which enabled the team to generate a nucleotide-free enzyme, and to design three respective defined nucleotide-conditions. The studies are described in detail and are state of the art. The resolution of the structures resolved, allow the interpretation of the data described in the manuscript with the focus of the three catalytic AB dimers during rotation. Such focus differs from the rotary studies of single molecule rotation studies, in which the rotary elements subunit γ and ϵ are in focus of the published studies.

The structures presented reveal, that the conformational changes in the three catalytic AB-dimers occur simultaneously, rather than in sequence, as has been shown for F-ATP synthases. The authors propose that rotation of the Tth V/A-ATPase proceeds via the tri-site model with the protein progressing through a two nucleotide bound state and a three nucleotide bound state. They also reveal that the unidirectional rotation occurs via a ratcheted-like mechanism driven by ATP cleavage instead of a power-stroke mechanism as described for F-ATPases. Finally, the data confirm that only 120° degree steps occur during rotation of the Tth V/A-ATPase as described for the A-type F-ATPase (Sielaff et al., *J. Biol. Chem.* (2016) Dec 2;291(49):25351-25363) and as in contrast to F-ATPases, for which substeps have been described. It is a pity that the detailed publication of Sielaff et al., which described in detail the

differences and similarities of archaeal A-ATP synthases, thermophilic and mesophilic ATP synthase molecular motors, is not cited and discussed in the manuscript.

In summary, the studies described in the manuscript by Kishikawa et al., are well designed and performed, and contribute to novel insights into the rotary mechanism of this molecular engine.

Our responses to Reviewer #1 comments,

We are very grateful to your detailed evaluation of our research and for your very positive comments. We have closely read the comments you have given to us and responded to each of them, as below. We hope that these responses will meet your requirements.

REVIEWER COMMENTS

Reviewer #1 (Remarks to the Author):

The manuscript by Kishikawa et al., provides significant new structural details on the V/A-ATPase and is an impressive study in terms of the data collected and the number of structures determined. This work will be significant in the field and should provide new insights into the V/A-ATPase and its comparison to other members of the field. With advances in cryoEM has come a depth of new information on rotary ATPases and their mechanism and this work will contribute to those studies. There are a few areas which it would be good to see considered and discussed within the manuscript;

How relevant are the concentrations of nucleotides used compared to what would be expected in the cell and how might differences influence the structure.

Response

The concentration of ATP in the cell has been reported to be on the order of millimolar [Imamura et al PNAS 2009 15651-6]. Therefore, the structure obtained at 6 mM ATP should represent the intracellular state. It is likely that the structure at 50 mM ATP does not exist in the cell. However, as mentioned in the text, the structure of the V/A-ATPases are almost unchanged by nucleotide binding, so it is likely that the determined structures in this study are relevant regardless of the higher ATP concentration found in the cell.

It is good to see a nanodisc being used to recreate a more native like membrane bilayer. Why is Vo more poorly resolved? Although a lower resolution why is the analysis of the different states in V1 not carried through into Vo. Are there any interesting observations in Vo which reflect the different nucleotide bound states?

Response

The lower resolution of the V_o part may be due to the fluctuation of the V_o domain relative to the V_1 domain. Indeed, the resolution of the DF subdomain connecting V_1 and V_o becomes lower towards the V_o side (see local resolution maps of *holo-V/A-ATPase* in Supplementary information). We expect that it is possible to obtain a high-resolution structure of the V_o by focusing on the V_o domain. We will report the results elsewhere if we obtain the V_o structures under various reaction conditions.

It would be good to see a broader analysis for what has been shown with the F and V ATPase family to expand on the impact.

Response

I agree with the reviewer's point. A similar analysis of FoF1 will be performed in the future and compared with our findings on V/A-ATPase in order to provide a broader context to the function of the rotary ATPases. Such a detailed analysis is not possible here.

Line 121, states 1 and 2 were improved, what about state 3?

Response

As the reviewer suggested the resolution of the V_1 domain of states 1 and 2 were improved using the focused refinement with signal subtraction techniques. However, in the case of state 3, the focused refinement did not work well. Because the signal subtraction was applied to a very small number of particles (about 5,000 particles), it is assumed that the refinement process did not converge. Since the structure of the V_1 domain in state 3 is expected to be similar to that in states 1 and 2, it does not affect the conclusions of our paper.

Line 151 although high resolution, $\sim 3\text{\AA}$ is not typically referred to as "atomic" resolution.

Response

Accordingly, we removed "atomic" and rewrote this sentence as follows:

Line 151

We identified a cryoEM structure of the original.....

Line 288, how likely is it that this system has 3 ATP's bound at any one time, how often would it naturally experience "saturation" concentrations?

Response

As mentioned earlier, since the cell is at a saturating concentration of ATP, this enzyme is likely to always have three nucleotides bound to the catalytic sites. In this state, the ATP binding dwell time is very short, therefore, the V/A-ATPase adopts V_{3nuc} structure at ATP saturated condition.

Line 306 demonstrates not demonstrate

Line 648, space after Cryo-EM

Response

We have modified the sentences as indicated.

Responses to Reviewer #2 comments,

We are very grateful for the appreciation of the significance of our research. In addition, we would like to thank you for your very helpful and positive comments. We believe that the comments we received have enhanced our research. The responses to each comment is given below in red. We hope that these responses will meet your requirements.

Reviewer #2 (Remarks to the Author):

Title: Structural snapshots of V/A-ATPase reveal a new paradigm for rotary catalysis

Journal: Nature Communications, NCOMMS-21-42328-T

All Authors: J. Kishikawa, A. Nakanishi, A. Nakano, S. Saeki, A. Furuta, T. Kato
K. Mistuoka, and K. Yokoyama

The primary source of ATP in archaea or the thermophilic bacterium, *Thermus thermophilus* (Tth) is the A-ATP synthase (in the manuscript called V/A-ATPase), which shares several structural features with eukaryotic V-ATPases, and is evolutionarily distant from the F-ATP synthases that fulfill this role in other bacteria and eukaryotes. Metabolism in archaea is coupled

to the generation of a H⁺- and/or Na⁺-potential across the membrane, both of which can provide the energy for ATP synthesis by the A-ATP synthase. Whereas the F-ATP synthases in prokaryotes and eukaryotes catalyze ATP synthesis at the expense of an electrochemical ion potential, the evolutionarily related V-ATPases function as ATP-driven ion pumps, and are unable to synthesize ATP under physiological condition. In recent years, the Yokoyama lab has contributed significant to the structural and mechanistic understanding of the Tth V/A-ATPase. In the study presented by Kishikawa et al., the authors describe the atomic cryo-EM structures of four different reactions conditions using the mutant enzyme A/S232A, T235S, which enabled the team to generate a nucleotide-free enzyme, and to design three respective defined nucleotide-conditions. The studies are described in detail and are state of the art. The resolution of the structures resolved, allow the interpretation of the data described in the manuscript with the focus of the three catalytic AB dimers during rotation. Such focus differs from the rotary studies of single molecule rotation studies, in which the rotary elements subunit γ and ϵ are in focus of the published studies.

The structures presented reveal, that the conformational changes in the three catalytic AB-dimers occur simultaneously, rather than in sequence, as has been shown for F-ATP synthases. The authors propose that rotation of the Tth V/A-ATPase proceeds via the tri-site model with the protein progressing through a two nucleotide bound state and a three nucleotide bound state. They also reveal that the unidirectional rotation occurs via a ratched-like mechanism driven by ATP cleavage instead of a power-stroke mechanism as described for F-ATPases. Finally, the data confirm that only 120° degree steps occur during rotation of the Tth V/A-ATPase as described for the A-type F-ATPase (Sielaff et al., J. Biol. Chem. (2016) Dec 2;291(49):25351-25363) and as in contrast to F-ATPases, for which substeps have been described. It is a pity that the detailed publication of Sielaff et al., which described in detail the differences and similarities of archaeal A-ATP synthases, thermophilic and mesophilic ATP synthase molecular motors, is not cited and discussed in the manuscript.

In summary, the studies described in the manuscript by Kishikawa et al., are well designed and performed, and contribute to novel insights into the rotary mechanism of this molecular engine.

Response

Thanks to the valuable comments from reviewer 2. By using a gold nano rod for high time resolution rotational experiments, Sielaff et al. showed that hydrolysis-driven rotational motion occurs after the catalytic dwell, consistent with the findings from our study. Therefore, we now cite this paper and have added the following sentence to the discussion.

Line 368

...are likely to share the same rotary mechanism. In fact, observation of rotation with high time resolved rotation analysis using a tiny gold rod showed that bacterial F-ATPase and an A-ATPase share the same rotation mechanism. For both rotary ATPases, a 120° rotation step together with ATP hydrolysis occurs after the catalytic dwell under ATP-saturated conditions.